# Recurrent loss of HMGCS2 shows that ketogenesis is not essential for the evolution of large mammalian brains

David Jebb[1,2,3], Michael Hiller[1,2,3]*

[1]Max Planck Institute of Molecular Cell Biology and Genetics, Dresden, Germany;
[2]Max Planck Institute for the Physics of Complex Systems, Dresden, Germany;
[3]Center for Systems Biology Dresden, Dresden, Germany

**Abstract** Apart from glucose, fatty acid-derived ketone bodies provide metabolic energy for the brain during fasting and neonatal development. We investigated the evolution of *HMGCS2*, the key enzyme required for ketone body biosynthesis (ketogenesis). Unexpectedly, we found that three mammalian lineages, comprising cetaceans (dolphins and whales), elephants and mastodons, and Old World fruit bats have lost this gene. Remarkably, many of these species have exceptionally large brains and signs of intelligent behavior. While fruit bats are sensitive to starvation, cetaceans and elephants can still withstand periods of fasting. This suggests that alternative strategies to fuel large brains during fasting evolved repeatedly and reveals flexibility in mammalian energy metabolism. Furthermore, we show that *HMGCS2* loss preceded brain size expansion in toothed whales and elephants. Thus, while ketogenesis was likely important for brain size expansion in modern humans, ketogenesis is not a universal precondition for the evolution of large mammalian brains.

DOI: https://doi.org/10.7554/eLife.38906.001

## Introduction

Periods of fasting are a common event for many animals (*Secor and Carey, 2016*). Fasting occurs due to natural food scarcity or as part of the life history strategy, for example during hibernation or migration. During fasting, the organism relies on stored sources of energy such as glucose in the form of glycogen and fatty acids (*Secor and Carey, 2016*). In addition, ketone bodies become an alternative fuel source that is important for many mammals to survive episodes of fasting or starvation (*Baird et al., 1972*; *Bouchat et al., 1981*; *Sicart et al., 1978*). For example, ketone bodies are used as an energy source in hibernating ground squirrels or elephant seal pups during their post-weaning fasting period (*Krilowicz, 1985*; *Castellini and Costa, 1990*). Notably, while the brain cannot metabolize fatty acids, ketone bodies can cross the blood-brain barrier and provide fuel under conditions of low blood glucose levels. For example, after starving for 3 days, the human brain takes 25% of its energy from ketone bodies and if fasting continues, ketone bodies replace glucose as the predominant fuel for brain metabolism (*Owen et al., 1967*; *Hasselbalch et al., 1994*). During the neonatal period, the developing human brain has high energy requirements and also relies on ketone bodies as a major fuel (*Cunnane and Crawford, 2003*; *Cahill, 2006*). Given their importance in fueling large, energetically expensive brains, it has been posited that ketone bodies do not only have an important role during fasting, but have also been crucial for brain expansion during human evolution (*Cunnane and Crawford, 2003*; *Wang et al., 2014*).

Ketone bodies comprise acetoacetate, acetone, and d-β-hydroxybutyrate (*Figure 1A*) and are mainly produced in the liver by ketogenesis. This metabolic process occurs in the mitochondria and uses fatty acid-derived acetyl-CoA to generate the water-soluble, acidic ketone bodies, which are

*For correspondence:
hiller@mpi-cbg.de

**Competing interests:** The authors declare that no competing interests exist.

**eLife digest** Our brain requires a lot of energy to work properly. Sugars are usually the main type of fuel for the body, but when they run low – for example during a food shortage – fat, in the form of fatty acids, can be used instead. However, the brain cannot directly process these molecules; instead, fatty acids need to go through ketogenesis, a process that turns fat into ketone bodies, which the organ can then burn. Scientists believe that the ability to create ketone bodies was essential for us to evolve large brains. Yet, it is still unclear if all mammals can transform fatty acids into ketone bodies. One way to look into this question is to track whether other species have HMGCS2, the main enzyme that drives ketogenesis.

Jebb and Hiller examined the genomes of 70 different species of mammals for the gene that codes for HMGCS2. The comparisons revealed that cetaceans (whales, dolphins and porpoises), Old World fruit bats and the African savanna elephant have all independently lost their working version of HMGCS2. Yet, many members of these three groups have evolved brains that are large for their body size. The genetic analyses showed that dolphins and elephants developed big brains after the enzyme became inactive, challenging the idea that HMGCS2 – and by extension ketogenesis – is always required for the evolution of large brains.

These results may also be useful for conservation efforts. Many fruit bats across the world are severely threatened, and their lack of ketogenesis could explain why these animals are highly sensitive to starvation and quickly die when food becomes scarce.

DOI: https://doi.org/10.7554/eLife.38906.002

secreted into the blood. The rate limiting step of ketogenesis is the production of 3-hydroxy-3-methylglutaryl-CoA (HMG-CoA) by HMG-CoA synthase (*Hegardt, 1999*). Mammals possess two HMG-CoA synthases that originated by gene duplication. While the cytosolic enzyme, encoded by *HMGCS1*, is broadly expressed and is necessary to produce cholesterol (*Hegardt, 1999*), the mitochondrial HMG-CoA synthase, encoded by *HMGCS2*, is primarily expressed in the liver and is only used for ketone body production. *HMGCS2* is required for ketogenesis, as mutations in the human gene and mouse gene-knockdown experiments abolish or greatly reduce ketogenesis (*Bouchard et al., 2001*; *Ramos et al., 2013*; *Thompson et al., 1997*; *Wolf et al., 2003*; *Pitt et al., 2015*; *Cotter et al., 2014*). HMG-CoA synthase-2 deficiency in human can lead to coma after fasting for more than 22 hours due to low glucose levels (*Thompson et al., 1997*; *Morris et al., 1998*). Human individuals with *HMGCS2* mutations therefore require regular carbohydrate intake but show no other symptoms, suggesting that this deficiency is probably underdiagnosed.

Here we investigated the evolution of *HMGCS2* in mammals. Unexpectedly, we identified three independent losses of this gene in cetaceans (dolphins and whales), pteropodids (Old World fruit-eating bats) and Elephantimorpha (elephants and mastodons). Remarkably, these species have relatively large brains, suggesting that, unlike in humans, ketone bodies are not strictly required for fueling complex brains. Furthermore, we show that in the cetacean and Elephantimorpha clades *HMGCS2* was lost before brain size expansion happened, suggesting that the lack of ketogenesis did not prohibit the evolution of large brains in these lineages. While strong conservation of *HMGCS2* in other mammals indicates that ketogenesis is a crucial metabolic process, the recurrent loss of this gene highlights an unexpected flexibility in mammalian energy metabolism.

## Results and discussion

To investigate the evolution of *HMGCS2*, we used a previously published whole genome alignment to inspect the gene sequence and the surrounding locus across 70 placental mammals (*Sharma and Hiller, 2017*). Surprisingly, we discovered that three independent lineages (cetaceans, pteropodids and the African savanna elephant) exhibit large deletions that remove *HMGCS2* exons or gene-inactivating mutations that shift the *HMGCS2* reading frame and destroy conserved splice site dinucleotides (*Figure 1B*). All three lineages have a deletion of exon one that encodes the mitochondrial targeting domain; such a deletion causes HMG-CoA synthase-2 deficiency in human individuals (*Pitt et al., 2015*). Other mutations affect exons encoding key residues required for HMG-CoA synthase catalytic activity and leave little of the coding sequence intact. Together with the deletion of

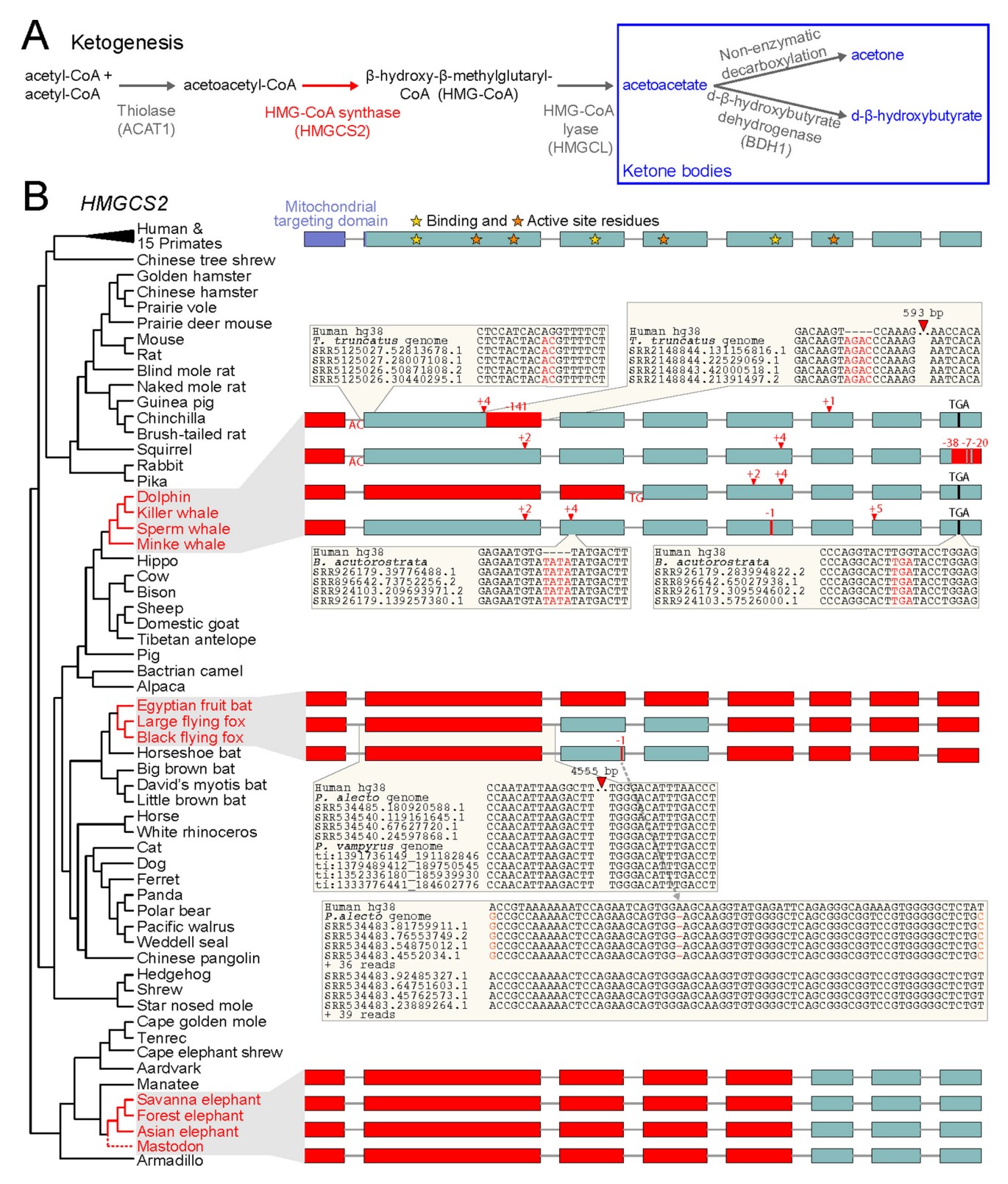

**Figure 1.** Evolution of ketogenesis in placental mammals. (**A**) Biosynthesis of ketone bodies (blue font). With the exception of the mitochondrial HMG-CoA synthase (HMGCS2, red font), the other two enzymes required for acetoacetate production also have roles in amino acid metabolism and are thus pleiotropic. BDH1 is only required for converting acetoacetate into d-β-hydroxybutyrate. (**B**) Recurrent loss of *HMGCS2* in three independent lineages (red font). All species in black font have an intact *HMGCS2* reading frame. Boxes are coding exons proportional to their size, introns are shown as

*Figure 1 continued on next page*

*Figure 1 continued*

horizontal lines. Red boxes are exon deletions. In-frame stop codon, frameshifting insertion/deletion and splice site-disrupting mutations are indicated. With the exception of the heterozygous 1 bp deletion in the black flying fox that has a read support of ~50:50 for the derived and ancestral allele and reveals two distinct haplotypes (inset), all shown mutations are supported by at least 30 reads with no support for the ancestral allele (***Supplementary file 1***). Insets exemplify the validation of inactivating mutations by showing the local genomic context and four reads.

DOI: https://doi.org/10.7554/eLife.38906.003

The following figure supplements are available for figure 1:

**Figure supplement 1.** Loss of the *HMGCS2* promoter region in the sperm whale, pteropodids, and the elephant.

DOI: https://doi.org/10.7554/eLife.38906.004

**Figure supplement 2.** Deletion of a large part of *HMGCS2* in elephant.

DOI: https://doi.org/10.7554/eLife.38906.005

**Figure supplement 3.** A shared ~ 270 bp deletion removed *HMGCS2* exon one in cetaceans.

DOI: https://doi.org/10.7554/eLife.38906.006

**Figure supplement 4.** Large deletions in the *HMGCS2* locus in pteropodid fruit bats.

DOI: https://doi.org/10.7554/eLife.38906.007

**Figure supplement 5.** Loss of the d-β-hydroxybutyrate producing *BDH1* gene.

DOI: https://doi.org/10.7554/eLife.38906.008

the promoter region in pteropodids, the elephant and the sperm whale (***Figure 1—figure supplement 1***), this shows that three mammalian lineages lost the enzyme that is required for ketogenesis.

In cetaceans and pteropodids, the remnants of the once-functional *HMGCS2* gene are located in a conserved genomic context with *REG4* upstream and *PHGDH* downstream. In elephant, the three remaining *HMGCS2* exons also occur in the same genomic locus adjacent to the conserved *PHGDH* gene, but inversions that already happened in the ancestor of elephants and the closely related manatees rearranged the locus upstream of *HMGCS2* (***Figure 1—figure supplement 2***). These rearrangements were succeeded by a large deletion in the elephant lineage that removed the first five *HMGCS2* exons together with the *REG4* gene.

To rule out that the gene-inactivating mutations are sequencing or genome assembly errors, we validated all smaller mutations and exon deletions with unassembled sequencing reads from the SRA and TRACE archives using blastn. All 22 mutations in cetaceans were confirmed by at least 30 reads, with no support for the non-gene-inactivating allele (***Figure 1B***, ***Supplementary file 1***). This includes the deletion of exon one that exhibits shared breakpoints in the toothed and baleen whale lineages (***Figure 1—figure supplement 3***), which strongly suggests that this deletion and thus *HMGCS2* loss already occurred before the split of the main cetacean lineages (***Figure 1C***). This is further supported by the 2 bp frameshifting insertion in exon two that is shared between killer whale and minke whale, and was later deleted in dolphin and sperm whale.

In pteropodid bats, the ~4.5 kb deletion that removed coding exon two is validated by sequencing reads and is shared between both flying foxes (***Figure 1B***), suggesting that *HMGCS2* was already lost in their common ancestor. Using the *HMGCS2* sequence of the David's myotis bat, we detected no evidence for the presence of the deleted *HMGCS2* exons in unassembled sequencing reads of both flying fox species, while we readily found all exons of the *HMGCS1* paralog, showing that the search is sufficiently sensitive. In the Egyptian fruit bat, *HMGCS2* is entirely removed by a large deletion between the *REG4* and *PHGDH* genes, which we validated with an independent Pac-Bio assembly (***Figure 1—figure supplement 4***). Consistent with ongoing gene erosion, the 1 bp deletion in exon three is heterozygous in the black flying fox (***Figure 1B***).

To rule out that the partial gene deletion in the African savanna elephant is an assembly error, we used the manatee *HMGCS2* sequence. Sensitive blastn searches found no significant hits for the deleted *HMGCS2* exons or the deleted neighboring *REG4* gene in the unassembled sequencing reads of two different savanna elephant individuals (***Cortez et al., 2014***). In contrast, the three remaining *HMGCS2* exons as well as all exons of the paralogous *HMGCS1* could be recovered. We also investigated related elephant species, making use of recently published sequence data from the African forest elephant and the Asian elephant (***Palkopoulou et al., 2018***; ***Reddy et al., 2015***). Further, we queried sequence data from two American mastodons, extinct Elephantimorpha that split from elephants 28–24 Mya (***Rohland et al., 2007***). As for the savanna elephant, the three remaining *HMGCS2* exons and entire *HMGCS1* gene were found in all three species, while the deleted

*HMGCS2* exons and the *REG4* gene were not found (*Figure 1B*). Parsimony suggests that the deletion, which removed large parts of *HMGCS2*, occurred prior to the divergence of mastodons and the elephant species.

We further found that the remaining *HMGCS2* sequence evolves under relaxed selection in cetaceans, pteropodids and Elephantimorpha (p<3e-3, *Supplementary file 2*). Together with the conserved genomic context, the lack of any evidence of a remaining functional *HMGCS2* in unassembled reads and the validated gene-inactivating mutations, we conclude that the main ketogenesis enzyme is lost in three independent mammalian lineages. Finally, we considered the possibility that HMGCS1, the cytosolic HMG-CoA synthase, may compensate for *HMGCS2* loss, which would require HMGCS1 to be localized in the mitochondria, where ketogenesis happens in other species. We found that the HMGCS1 protein of cetaceans, pteropodids and elephant does not possess a mitochondrial targeting domain. Furthermore, an analysis of available liver RNA-seq data from the minke whale and Egyptian fruit bat provides no indication of alternative or novel exons in *HMGCS1* that could encode such a targeting signal. Thus, HMGCS1 does not seem to be capable of compensating for the loss of *HMGCS2*, suggesting that ketogenesis is lost in cetaceans, pteropodids and Elephantimorpha.

Next, we investigated whether the loss of *HMGCS2* is associated with the loss of other enzymes in the ketogenesis pathway (*Figure 1A*). *ACAT1* and *HMGCL* do not exhibit inactivating mutations in cetaceans, pteropodids and the elephant, likely because the respective enzymes are not only required for the production of ketone bodies but are also involved in leucine and isoleucine metabolism. In contrast to these two pleiotropic genes, *BDH1* is only involved in converting acetoacetate into the ketone body d-β-hydroxybutyrate (*Figure 1A*). We found that *BDH1* exhibits several inactivating mutations and evolved under relaxed selection in cetaceans and pteropodids (*Figure 1—figure supplement 5*, *Supplementary file 2*). Overall, this suggests that the loss of *HMGCS2* is only associated with the loss of non-pleiotropic genes in the ketogenesis pathway.

The 59 other mammals, for which the genome assembly fully covered the *HMGCS2* locus (*Figure 1B*), do not exhibit inactivating mutations in this gene. Consistent with the presence of a functional gene, we further estimated an average non-synonymous/synonymous (dN/dS) ratio of 0.16, which indicates that *HMGCS2* evolves under strong purifying selection in other mammals.

The observation that *HMGCS2* is well-conserved in the majority of mammals is consistent with ketogenesis being an important metabolic process. However, the recurrent loss of *HMGCS2* raises the question of which energy source is used by the brain during fasting. Consistent with the loss of ketogenesis in cetaceans, bottlenose dolphins do not produce ketone bodies after fasting for 3 days but are nevertheless able to maintain high blood glucose levels over this entire period (*Ridgway, 2013*). It was suggested that dolphins maintain high glucose levels by synthesis of glucose from non-carbohydrates (gluconeogenesis), in particular from glucogenic amino acids that are abundant in their diet (*Ridgway, 2013*). This suggests that ketogenesis became dispensable in dolphins and that *HMGCS2* was lost as a consequence of relaxed or no selection to maintain this gene. Similarly, the loss of ketogenesis in pteropodid fruit bats may be a consequence of the relatively constant availability of fruit year-round, which provides large quantities of glucose. This is in agreement with molecular dating, which estimates that the loss of *HMGCS2* happened rather late in the lineage leading to the fruit bat clade and may even have occurred independently after the split of the frugivorous flying foxes and the Egyptian fruit bat (*Figure 2A*). Consistent with lack of ketone bodies as alternative fuel, Egyptian fruit bats that were fasted for more than 24 hours in captivity frequently died (*van der Westhuyzen, 1978*). Thus, like HMG-CoA synthase-2 deficient human individuals, these bats are sensitive to starvation. Hence, while ketogenesis may have been lost under ancestral conditions of constantly available, glucose-rich food, the loss of *HMGCS2* may now represent a disadvantage, which will be of interest to ongoing conservation efforts for ecologically and economically important species in the pteropodid family. In contrast to cetaceans and fruit bats, little is known about how elephants respond to fasting; however, the following observation is consistent with the loss of ketogenesis. During musth, when elephant males experience longer periods of fasting and can lose 10% of their body weight, their blood becomes slightly more alkaline (*Rasmussen and Perrin, 1999*). This is contrary to an increased blood acidity that would be expected from an increasing production of acidic ketone bodies.

Given the importance of ketogenesis to provide energy to the brain during starvation, it is noteworthy that species in all three *HMGCS2*-loss lineages generally have large relative brain sizes

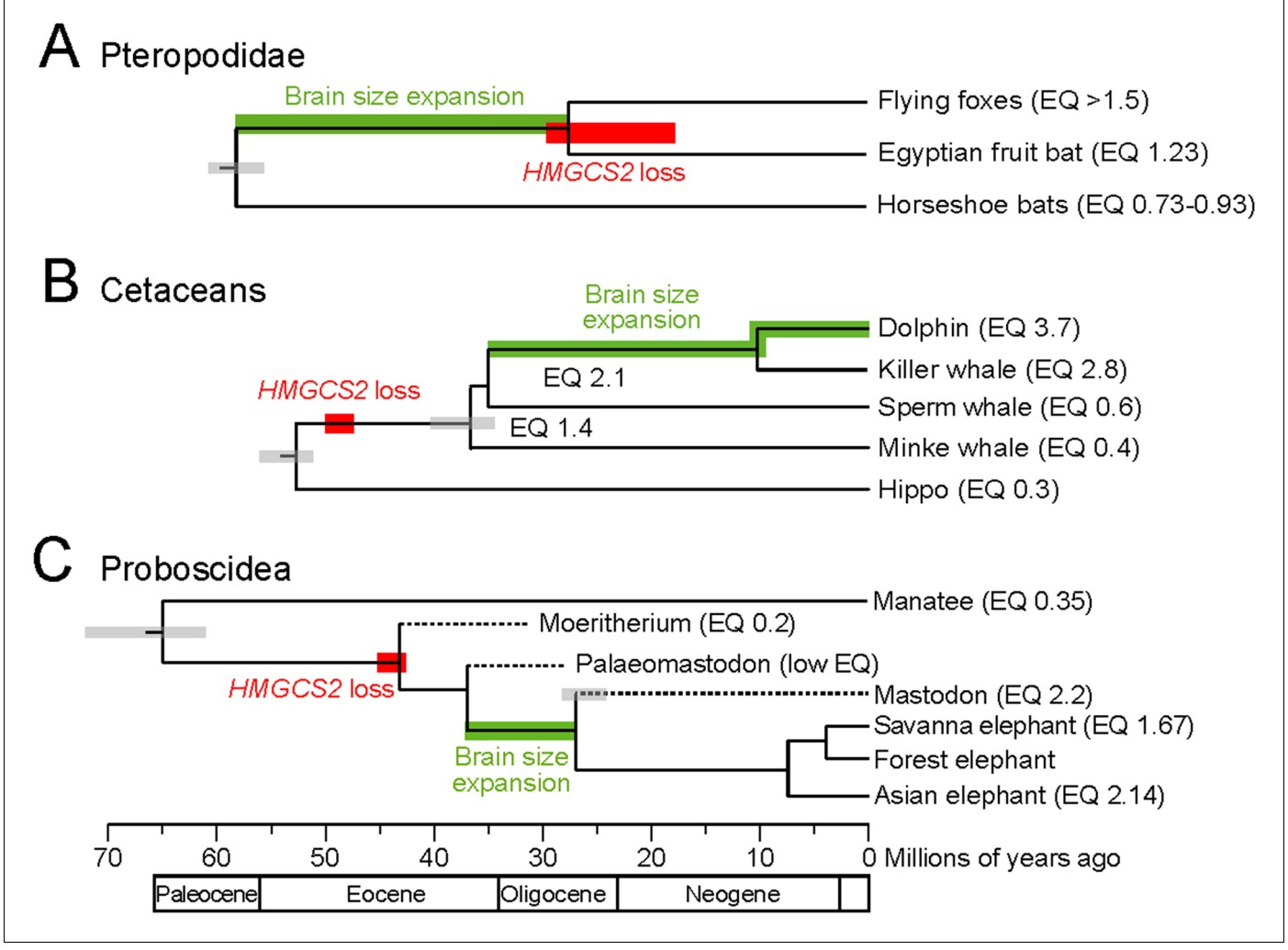

**Figure 2.** *HMGCS2* loss and brain size evolution. (**A**) In pteropodids, molecular dating estimates that the loss of *HMGCS2* happened 29–18 Mya and thus may overlap the split of the flying foxes and the Egyptian fruit bat. It is not possible to resolve whether gene loss happened before or after the split as *HMGCS2* is completely deleted in the Egyptian fruit bat. While horseshoe bats and other insectivorous bat lineages have brains not larger than expected for their body size (encephalization quotient (EQ) <1), brain size has increased in the lineage leading to the fruit bats that have EQ values > 1 (***Stephan et al., 1981***). Thus, brain size expansion presumably predates the loss of ketogenesis. (**B**) *HMGCS2* was already lost in the cetacean ancestor before the split of toothed and baleen whales ~ 36 Mya, as inferred from shared inactivating mutations in exons 1, 2 and 8. Molecular dating further estimates that the loss of this gene happened early on the cetacean branch 50–47 Mya. The cetacean ancestor had a brain slightly larger than expected for its body size with an EQ of 1.4. While EQ values increased and decreased in several cetacean lineages, brain size has greatly expanded in dolphins, reaching an EQ of 3.7 (***Montgomery et al., 2013***). Thus, brain size expansion in dolphins occurred after the loss of ketogenesis. (**C**) Early proboscids such as *Moeritherium*, an extinct lineage that split from other proboscids ~ 43 Mya, had brains about 20% of the size expected for a mammal of the same body size, and thus an EQ of 0.2 (***Shoshani et al., 2006***). Exact EQ values of Palaeomastodons are not known; however, fossils have a small braincase, which indicates a low EQ (***Sanders et al., 2010***; ***Benoit, 2015***). In contrast, mastodons that diverged from elephants ~ 27 Mya had brains about twice as large as expected from their body size (EQ 2.2), similar to extant elephants (***Shoshani et al., 2006***). This suggests that brain size expansion happened in a period between 37 and 27 Mya. Molecular dating indicates that *HMGCS2* loss happened between 45 and 42 Mya, suggesting that the loss of ketogenesis precedes brain size expansion in the elephant lineage. Divergence times of extinct proboscid lineages were taken from (***Shoshani and Tassy, 2013***) and (***Rohland et al., 2007***). Supporting Information.

DOI: https://doi.org/10.7554/eLife.38906.009

The following source data is available for figure 2:

**Source data 1.** Sequence alignment.

DOI: https://doi.org/10.7554/eLife.38906.010

(*Stephan et al., 1981*; *Boddy et al., 2012*). For example, the encephalization quotient (EQ), measuring the ratio between the observed brain size and the size expected for a mammal of the same body weight, is 3.7 for the bottlenose dolphin (*Montgomery et al., 2013*). Compared to human, dolphins and elephants are also among the few mammals that have a higher degree of neocortex folding, a measure that positively correlates with neuron number (*Manger et al., 2012*; *Lewitus et al., 2014*). Furthermore, while powered flight imposes a constraint on body and brain size in bats, pteropodid fruit bats exhibit a well-developed visual brain system and have brains nearly twice as large as that of insectivorous vesper bats of equal body weight (*Stephan et al., 1981*). Species in all three lineages also exhibit cognitive behaviors that are regarded as a sign of intelligence, exemplified by vocal learning and, in dolphins and elephants, by complex social structures, tool use and self-recognition (*Krützen et al., 2005*; *Foerder et al., 2011*; *Poole et al., 2005*; *Prat et al., 2015*; *Plotnik et al., 2006*). Thus, the loss of *HMGCS2* in independent large-brained species suggests that ketone bodies are not strictly required to fuel large mammalian brains during fasting.

Finally, the timing of *HMGCS2* loss has implications for understanding the general preconditions for brain size expansion during the evolution of mammals. While the loss of *HMGCS2* in pteropodids likely happened after brain size expansion in this lineage (*Figure 2A*), shared inactivating mutations show that *HMGCS2* was already inactivated in the cetacean ancestor, and thus prior to a period of brain size expansion that resulted in the large brains of dolphins (*Boddy et al., 2012*; *Montgomery et al., 2013*) (*Figure 2B*). For the elephant lineage, we used molecular dating to estimate that *HMGCS2* was lost around 45–42 Mya (*Supplementary file 3*). Thus, like in toothed whales, the loss of this gene likely occurred prior to the period that led to large relative brain sizes in modern elephants (*Shoshani et al., 2006*) (*Figure 2C*). Consequently, while ketogenesis was likely a crucial factor for brain size increase in humans (*Cunnane and Crawford, 2003*; *Wang et al., 2014*), the loss of ketogenesis has not prohibited drastic evolutionary brain size expansion in two other mammalian lineages.

In conclusion, we have identified three independent losses of *HMGCS2* in placental mammals. While this may contribute to starvation sensitivity in fruit bats, cetaceans and elephants can withstand periods of fasting. Hence, alternative strategies to fuel large brains during fasting have evolved at least twice, revealing flexibility in the energy metabolism of mammals. Finally, the timing of *HMGCS2* loss indicates that ketogenesis is not a universal precondition for the evolution of large mammalian brains. More generally, our results further highlight the potential of comparative gene analyses (*Emerling and Springer, 2014*; *Meredith et al., 2009*; *Castro et al., 2014*; *Albalat and Cañestro, 2016*; *Lopes-Marques et al., 2017*; *Hecker et al., 2017*; *Gaudry et al., 2017*; *Sharma et al., 2018a*; *Sharma et al., 2018b*; *Meyer et al., 2018*; *Emerling et al., 2018*) to reveal novel insights into the evolution of metabolic, physiological or morphological phenotypes.

## Materials and methods

### Key resources table

| Reagent type (species) or resource | Designation | Source or reference | Identifiers | Additional information |
|---|---|---|---|---|
| Software, algorithm | CESAR | https://github.com/hillerlab/CESAR2.0 | | |
| Software, algorithm | Lastz (1.03.54) | http://www.bx.psu.edu/~rsharris/lastz/newer/lastz-1.03.54.tar.gz | | K = 2400 L = 3000 |
| Software, algorithm | axtChain | https://github.com/ucsc GenomeBrowser/kent | | |
| Software, algorithm | chainNet | https://github.com/ucsc GenomeBrowser/kent | | |
| Software, algorithm | BLAST (2.6.0+) | ftp://ftp.ncbi.nlm.nih.gov/blast/executables/blast+/LATEST/ncbi-blast-2.6.0+-x64-linux.tar.gz | RRID:SCR_004870 | word_size = 7 |

*Continued on next page*

*Continued*

| Reagent type (species) or resource | Designation | Source or reference | Identifiers | Additional information |
|---|---|---|---|---|
| Software, algorithm | MACSE (2.01) | https://bioweb.supagro.inra.fr/macse/ | | -prog enrichAlignment -prog refineAlignment |
| Software, algorithm | RELAX (HYPHY 2.3.11) | https://github.com/veg/hyphy | RRID:SCR_016162 | |
| Software, algorithm | PAML (4.0) | http://abacus.gene.ucl.ac.uk/software/paml.html | RRID:SCR_014932 | |
| Software, algorithm | TargetP (1.1) | http://www.cbs.dtu.dk/services/TargetP/ | | |
| Software, algorithm | HISAT2 (2.0.0) | https://ccb.jhu.edu/software/hisat2/index.shtml | RRID:SCR_015530 | |
| Software, algorithm | SAMtools (1.1) | https://github.com/samtools/samtools | RRID:SCR_002105 | |

## Investigating the *HMGCS2* reading frame

To investigate the *HMGCS2* sequence across mammals, we used a whole genome alignment between the human reference genome (hg38 assembly) and 69 other placental mammal genomes (*Sharma and Hiller, 2017*). In addition to these assemblies, we downloaded the genome assembly of the Hippopotamus (*Árnason et al., 2018*) (NCBI GCA_002995585.1) and updated genome assemblies of the Large flying fox (NCBI GCF_000151845.1), the Egyptian fruit bat (NCBI GCF_001466805.2) and the African savanna elephant (ftp://ftp.broadinstitute.org/distribution/assemblies/mammals/elephant/loxAfr4/). For these four assemblies, we computed pairwise alignment chains to the human hg38 genome by applying lastz (*Harris, 2007*) with parameters K = 2400, L = 3000 and the default scoring matrix, axtChain (*Kent et al., 2003*) and chainCleaner (*Suarez et al., 2017*) (both with default parameters). Collinear alignment chains were visualized in the UCSC genome browser (*Casper et al., 2018*) and inspected for conserved synteny with adjacent genes. All analyzed genome assemblies are listed in *Supplementary file 4*.

We used the gene loss detection approach (*Sharma et al., 2018a*) to search across all mammals for mutations that could inactivate *HMGCS2*. This approach considers large deletions that cover exons, frameshifting insertions and deletions, mutations that disrupt donor (GT/GC) or acceptor (AG) splice site dinucleotides, and nonsense mutations. To exclude false inactivating mutations caused by alignment ambiguities, this method only considers those putative inactivating mutations that were confirmed by CESAR (*Sharma et al., 2016*; *Sharma et al., 2017*), a method trained to output an intact exon alignment whenever possible. Furthermore, exon deletions or exonic regions that do not align between human and another species were only considered if the respective locus did not overlap an assembly gap in the other genome (*Hiller et al., 2012*). For the proboscis monkey and lesser Egyptian jerboa, greater than 20% of the *HMGCS2* protein-coding region was ambiguous bases due to assembly gaps. These species were classified as 'missing', as it is not possible to unambiguously determine presence or absence of *HMGCS2*.

## Validation of gene-inactivating mutations

Exon losses and inactivating mutations identified were manually validated using unassembled sequencing read data from the TRACE and Sequence Read Archives. To validate exon losses, we used sensitive blastn runs (word size = 7) to search read data of *HMGCS2* loss species. As queries, we used *HMGSC1* and *HMGCS2* exon sequences from a closely related species with an intact *HMGCS2* gene. Specifically, we used the cow sequence to search cetacean read data, and the sequence of David's myotis bat to search pteropodid read data. Read data from elephants and mastodon was searched using the manatee *HMGSC1*, *HMGCS2* and *REG4* exonic sequence. To validate smaller inactivating mutations (stop codon, frameshift and splice site mutations) and exon deletions, we extracted the genomic context 50 bp up- and downstream of each inactivating mutation in an *HMGCS2* loss species and determined the number of sequencing reads that support the derived

(inactivating) and ancestral (non-inactivating) allele, as described in (*Hecker et al., 2017*). SRA accessions are provided in *Supplementary file 5*.

## Relaxed selection analysis

We generated a multiple sequence alignment of the *HMGCS2* coding sequence from the CESAR alignments and replaced in-frame stop codons with 'NNN'. Using MACSE v2 (*Ranwez et al., 2018*), we added to this alignment the Chinese Horseshoe bat (*Rhinolophus sinicus*, XM_019730577) and the *Hippopotamus amphibius HMGCS2* coding sequence as well as the inferred exonic sequences of the Asian elephant, the African forest elephant and the mastodon. The alignment was then refined using MACSE v2 prior to visual inspection and further refinement. RELAX (*Wertheim et al., 2015*) was applied to test for relaxation of selection. First, we designated all branches within the cetacean, pteropodid and elephant/mastodon subtrees as foreground and designated all other branches as background. Second, we tested each subtree separately against the background branches, removing the other two *HMGCS2* loss lineages. We also tested the elephant lineage including only the African savanna elephant.

## Molecular dating of *HMGCS2* loss in the elephant lineage

To date the loss of *HMGCS2* along the putative loss branches in the phylogenetic tree, we used the method described in (*Meredith et al., 2009*; *Gaudry et al., 2017*), which estimates the portion of the loss branch where the gene evolved under selection and the portion where it evolved neutrally. Since synonymous positions do not entirely evolve neutrally due to constraints on splicing and translation, this approach assumes that the synonymous mutation rate of a functional gene is 70% of the fully-neutral synonymous mutation rate of an inactivated gene. Upper and lower bounds of species divergence times, the estimated length of the loss branch and respective sources are given in *Supplementary file 3*. The branch model in PAML (*Yang, 2007*) was fit, with five dN/dS classes, one for each of the three loss branches, one for the subsequent pseudogene branches and a final class for all functional branches. Pseudogene branches were assumed to evolve with a dN/dS of 1 for the dating calculations. We also fit models for each loss lineage individually and further tested the elephant lineage including only the African savanna elephant.

## Investigating the possibility of co-option of *HMGCS1*

We tested the amino acid sequences of the annotated or CESAR-inferred HMGCS1 protein from all *HMGCS2*-loss species for the presence of a potential mitochondrial target peptide (mTP) using TargetP (*Emanuelsson et al., 2007*). This revealed no evidence for the presence of an mTP in any species. To investigate the possibility that an mTP is provided by a novel or alternative first coding exon, we inspected gene predictions from Augustus that were available for all species. Those predicted gene models that contained an alternative first exon were found to not have an mTP. Furthermore, we used RNA-seq data from liver, the primary site of ketogenesis in other species, which was available for the Egyptian fruit bat (SRA SRR2914059, SRR2914369) and the minke whale (SRR919296). RNA-seq reads were mapped to the genome using HISAT2 (*Kim et al., 2015*), SAM files were sorted and converted to BAM files using SAMtools (*Li et al., 2009*) prior to visualization in the UCSC genome browser. For both species, we found no evidence of alternative or novel exons that could result in a different HMGCS1 N-terminus.

## Investigating the loss of other ketogenesis enzymes

Three other genes, *ACAT1*, *HMGCL* and *BDH1*, which encode components of the ketogenesis pathway were investigated for potential inactivating mutations using the same gene loss pipeline and mutation validation strategy described above. These genes were also tested for signs of relaxed selection in the three *HMGCS2*-loss lineages using RELAX (*Supplementary file 2*).

## Data availability

All data analyzed during this study is publicly available on NCBI, SRA and the Trace Archive. The multiple sequence alignment of the mammalian *HMGCS2* coding sequences is provided as *Figure 2—source data 1*.

## Acknowledgements

We thank Martin Pippel, Gene Myers, Sonja Vernes and Emma Teeling for access to PacBio reads of Rousettus aegyptiacus. We also thank Nikolai Hecker for computing genome alignments and source code for read validation, Sider Penkov for helpful discussions, Juliana Roscito for comments on the manuscript and the Computer Service Facilities of the MPI-CBG and MPI-PKS for their support. This work was supported by the Max Planck Society and the German Research Foundation (HI 1423/3–1).

## Additional information

### Funding

| Funder | Grant reference number | Author |
|---|---|---|
| Max-Planck-Gesellschaft | Open-access funding | Michael Hiller |
| Deutsche Forschungsgemeinschaft | HI 1423/3-1 | Michael Hiller |

The funders had no role in study design, data collection and interpretation, or the decision to submit the work for publication.

### Author contributions

David Jebb, Formal analysis, Validation, Investigation, Visualization, Methodology, Writing—original draft, Writing—review and editing; Michael Hiller, Conceptualization, Supervision, Funding acquisition, Investigation, Visualization, Writing—original draft, Project administration, Writing—review and editing

### Author ORCIDs

David Jebb http://orcid.org/0000-0001-6362-7378
Michael Hiller http://orcid.org/0000-0003-3024-1449

### Decision letter and Author response

Decision letter https://doi.org/10.7554/eLife.38906.088
Author response https://doi.org/10.7554/eLife.38906.089

## Additional files

### Supplementary files

• Supplementary file 1. Number of unassembled sequence reads supporting gene-inactivating mutations in cetaceans and fruit bats. Please note that sequencing errors can change a real stop codon to a sense codon, which happened in two cases; however, given that > 60 reads confirm the stop codon, these single erroneous reads to not support the presence of the ancestral allele. The heterozygous 1 bp deletion in the black flying fox (last row) is discussed in the main text.
DOI: https://doi.org/10.7554/eLife.38906.011

• Supplementary file 2. RELAX analysis of genes in the ketogenesis pathway. The estimate for 'All' and Elephantimorpha was computed from an alignment including the Asian elephant, the African forest elephant and the mastodon.
DOI: https://doi.org/10.7554/eLife.38906.012

• Supplementary file 3. Dating the loss of *HMGCS2*. The table lists divergence times and estimates for how long *HMGCS2* was functional along the mixed branch leading to the three loss lineages. Using the method of (*Meredith et al., 2009*; *Gaudry et al., 2017*), we calculated a point estimate for when *HMGCS2* was lost and an upper/lower bound of this estimate. (A) Elephant lineage. We estimated *HMGCS2* loss dates both from an alignment that includes sequences of Elephantimorpha and from an alignment that includes only the African savanna elephant. (B) Fruit bat lineage. (C) Cetacean lineage.
DOI: https://doi.org/10.7554/eLife.38906.013

• Supplementary file 4. Species and genome assemblies that were analyzed in this study.
DOI: https://doi.org/10.7554/eLife.38906.014

• Supplementary file 5. Sources of unassembled genomic sequencing reads used for the validation of inactivating mutations in *HMGCS2* loss species
DOI: https://doi.org/10.7554/eLife.38906.015

• Transparent reporting form
DOI: https://doi.org/10.7554/eLife.38906.016

## Data availability

All data analyzed during this study are publicly available on NCBI, SRA and the Trace Archive. The alignment used to date gene loss provided as a source file.

The following previously published datasets were used:

| Author(s) | Year | Dataset title | Dataset URL | Database, license, and accessibility information |
|---|---|---|---|---|
| Zhang G, Cowled C, Shi Z, Huang Z, Bishop-Lilly KA, Fang X, Wynne JW, Xiong Z, Baker ML, Zhao W, Tachedjian M, Zhu Y, Zhou P, Jiang X, Ng J, Yang L, Wu L, Xiao J, Feng Y | 2013 | Black flying fox raw sequence reads | https://www.ncbi.nlm.nih.gov/sra/SRX174444 | Publicly available at the NCBI Sequence Read Archive (accession no. SRX174444) |
| Zhang G, Cowled C, Shi Z, Huang Z, Bishop-Lilly KA, Fang X, Wynne JW, Xiong Z, Baker ML, Zhao W, Tachedjian M, Zhu Y, Zhou P, Jiang X, Ng J, Yang L, Wu L, Xiao J, Feng Y, Chen Y, Sun X, Zhang Y, Marsh GA, Crameri G, Broder CC, Frey KG, Wang LF, Wang J. | 2013 | Black flying fox raw sequence reads | https://www.ncbi.nlm.nih.gov/sra/SRX174445 | Publicly available at the NCBI Sequence Read Archive (accession no. SRX174445) |
| Zhang G, Cowled C, Shi Z, Huang Z, Bishop-Lilly KA, Fang X, Wynne JW, Xiong Z, Baker ML, Zhao W, Tachedjian M, Zhu Y, Zhou P, Jiang X, Ng J, Yang L, Wu L, Xiao J, Feng Y, Chen Y, Sun X, Zhang Y, Marsh GA, Crameri G, Broder CC, Frey KG, Wang LF, Wang J. | 2013 | Black flying fox raw sequence reads | https://www.ncbi.nlm.nih.gov/sra/SRX174446 | Publicly available at the NCBI Sequence Read Archive (accession no. SRX174446 |
| Zhang G, Cowled C, Shi Z, Huang Z, Bishop-Lilly KA, Fang X, Wynne JW, Xiong Z, Baker ML, Zhao W, Tachedjian M, Zhu Y, Zhou P, Jiang X, Ng J, Yang L, Wu L, Xiao J, Feng Y, Chen Y, Sun X, Zhang Y, | 2013 | Black flying fox raw sequence reads | https://www.ncbi.nlm.nih.gov/sra/SRX174448 | Publicly available at the NCBI Sequence Read Archive (accession no. SRX174448) |

| | | | | |
|---|---|---|---|---|
| Marsh GA, Crameri G, Broder CC, Frey KG, Wang LF, Wang J. | | | | |
| Kerstin Lindblad-Toh, Manuel Garber, Or Zuk, Michael F. Lin, Brian J. Parker, Stefan Washietl, Pouya Kheradpour, Jason Ernst, Gregory Jordan, Evan Mauceli, Lucas D. Ward, Craig B. Lowe, Alisha K. Holloway, Michele Clamp, Sante Gnerre, Jessica Alföldi, Kathryn Beal, Jean Chang, Hiram Clawson, James Cuff, Federica Di Palma, Stephen Fitzgerald, Paul Flicek, Mitchell Guttman, Melissa J. Hubisz, David B. Jaffe, Irwin Jungreis, W. James Kent, Dennis Kostka, Marcia Lara, Andre L. Martins, Tim Massingham, Ida Moltke, Brian J. Raney, Matthew D. Rasmussen, Jim Robinson, Alexander Stark, Albert J. Vilella, Jiayu Wen, Xiaohui Xie, Michael C. Zody, Broad Institute Sequencing Platform and Whole Genome Assembly Team, Kim C. Worley, Christie L. Kovar, Donna M. Muzny, Richard A. Gibbs, Baylor College of Medicine Human Genome Sequencing Center Sequencing Team, Wesley C. Warren, Elaine R. Mardis, George M. Weinstock, Richard K. Wilson, Genome Institute at Washington University, Ewan Birney, Elliott H. Margulies, Javier Herrero, Eric D. Green, David Haussler, Adam Siepel, Nick Goldman, Katherine S. Pollard, Jakob S. Pedersen, Eric S. Lander & Manolis Kellis | 2011 | Large flying fox raw sequence reads | https://www.hgsc.bcm.edu/other-mammals/megabat-genome-project | Publicly available at the Human Genome Sequencing Center |
| Kerstin Lindblad- | 2011 | Large flying fox raw sequence reads | https://www.ncbi.nlm. | Publicly available at |

| | | | | |
|---|---|---|---|---|
| Toh, Manuel Garber, Or Zuk, Michael F. Lin, Brian J. Parker, Stefan Washietl, Pouya Kheradpour, Jason Ernst, Gregory Jordan, Evan Mauceli, Lucas D. Ward, Craig B. Lowe, Alisha K. Holloway, Michele Clamp, Sante Gnerre, Jessica Alföldi, Kathryn Beal, Jean Chang, Hiram Clawson, James Cuff, Federica Di Palma, Stephen Fitzgerald, Paul Flicek, Mitchell Guttman, Melissa J. Hubisz, David B. Jaffe, Irwin Jungreis, W. James Kent, Dennis Kostka, Marcia Lara, Andre L. Martins, Tim Massingham, Ida Moltke, Brian J. Raney, Matthew D. Rasmussen, Jim Robinson, Alexander Stark, Albert J. Vilella, Jiayu Wen, Xiaohui Xie, Michael C. Zody, Broad Institute Sequencing Platform and Whole Genome Assembly Team, Kim C. Worley, Christie L. Kovar, Donna M. Muzny, Richard A. Gibbs, Baylor College of Medicine Human Genome Sequencing Center Sequencing Team, Wesley C. Warren, Elaine R. Mardis, George M. Weinstock, Richard K. Wilson, Genome Institute at Washington University, Ewan Birney, Elliott H. Margulies, Javier Herrero, Eric D. Green, David Haussler, Adam Siepel, Nick Goldman, Katherine S. Pollard, Jakob S. Pedersen, Eric S. Lander & Manolis Kellis | | | nih.gov/sra/SRX708163 | the NCBI Sequence Read Archive (accession no. SRX708163) |
| Kerstin Lindblad-Toh, Manuel Garber, Or Zuk, Michael F. Lin, Brian J. Parker, Stefan Washietl, Pouya Kher- | 2011 | Large flying fox raw sequence reads | https://www.ncbi.nlm.nih.gov/sra/SRX708167 | Publicly available at the NCBI Sequence Read Archive (accession no. SRX708167) |

| | | | | |
|---|---|---|---|---|
| adpour, Jason Ernst, Gregory Jordan, Evan Mauceli, Lucas D. Ward, Craig B. Lowe, Alisha K. Holloway, Michele Clamp, Sante Gnerre, Jessica Alföldi, Kathryn Beal, Jean Chang, Hiram Clawson, James Cuff, Federica Di Palma, Stephen Fitzgerald, Paul Flicek, Mitchell Guttman, Melissa J. Hubisz, David B. Jaffe, Irwin Jungreis, W. James Kent, Dennis Kostka, Marcia Lara, Andre L. Martins, Tim Massingham, Ida Moltke, Brian J. Raney, Matthew D. Rasmussen, Jim Robinson, Alexander Stark, Albert J. Vilella, Jiayu Wen, Xiaohui Xie, Michael C. Zody, Broad Institute Sequencing Platform and Whole Genome Assembly Team, Kim C. Worley, Christie L. Kovar, Donna M. Muzny, Richard A. Gibbs, Baylor College of Medicine Human Genome Sequencing Center Sequencing Team, Wesley C. Warren, Elaine R. Mardis, George M. Weinstock, Richard K. Wilson, Genome Institute at Washington University, Ewan Birney, Elliott H. Margulies, Javier Herrero, Eric D. Green, David Haussler, Adam Siepel, Nick Goldman, Katherine S. Pollard, Jakob S. Pedersen, Eric S. Lander & Manolis Kellis | | | | |
| Kerstin Lindblad-Toh, Manuel Garber, Or Zuk, Michael F. Lin, Brian J. Parker, Stefan Washietl, Pouya Kheradpour, Jason Ernst, Gregory Jordan, Evan Mauceli, Lucas D. Ward, Craig B. Lowe, | 2011 | Large flying fox raw sequence reads | https://www.ncbi.nlm.nih.gov/sra/SRX708168 | Publicly available at the NCBI Sequence Read archive (accession no. SRX708168) |

| | | | | |
|---|---|---|---|---|
| Alisha K. Holloway, Michele Clamp, Sante Gnerre, Jessica Alföldi, Kathryn Beal, Jean Chang, Hiram Clawson, James Cuff, Federica Di Palma, Stephen Fitzgerald, Paul Flicek, Mitchell Guttman, Melissa J. Hubisz, David B. Jaffe, Irwin Jungreis, W. James Kent, Dennis Kostka, Marcia Lara, Andre L. Martins, Tim Massingham, Ida Moltke, Brian J. Raney, Matthew D. Rasmussen, Jim Robinson, Alexander Stark, Albert J. Vilella, Jiayu Wen, Xiaohui Xie, Michael C. Zody, Broad Institute Sequencing Platform and Whole Genome Assembly Team, Kim C. Worley, Christie L. Kovar, Donna M. Muzny, Richard A. Gibbs, Baylor College of Medicine Human Genome Sequencing Center Sequencing Team, Wesley C. Warren, Elaine R. Mardis, George M. Weinstock, Richard K. Wilson, Genome Institute at Washington University, Ewan Birney, Elliott H. Margulies, Javier Herrero, Eric D. Green, David Haussler, Adam Siepel, Nick Goldman, Katherine S. Pollard, Jakob S. Pedersen, Eric S. Lander & Manolis Kellis | | | | |
| Kerstin Lindblad-Toh, Manuel Garber, Or Zuk, Michael F. Lin, Brian J. Parker, Stefan Washietl, Pouya Kheradpour, Jason Ernst, Gregory Jordan, Evan Mauceli, Lucas D. Ward, Craig B. Lowe, Alisha K. Holloway, Michele Clamp, Sante Gnerre, Jessica Alföldi, Kathryn Beal, Jean Chang, | 2011 | Large flying fox raw sequence reads | https://www.ncbi.nlm.nih.gov/sra/SRX708169 | Publicly available at the NCBI Sequence Read archive (accession no. SRX708169) |

Hiram Clawson, James Cuff, Federica Di Palma, Stephen Fitzgerald, Paul Flicek, Mitchell Guttman, Melissa J. Hubisz, David B. Jaffe, Irwin Jungreis, W. James Kent, Dennis Kostka, Marcia Lara, Andre L. Martins, Tim Massingham, Ida Moltke, Brian J. Raney, Matthew D. Rasmussen, Jim Robinson, Alexander Stark, Albert J. Vilella, Jiayu Wen, Xiaohui Xie, Michael C. Zody, Broad Institute Sequencing Platform and Whole Genome Assembly Team, Kim C. Worley, Christie L. Kovar, Donna M. Muzny, Richard A. Gibbs, Baylor College of Medicine Human Genome Sequencing Center Sequencing Team, Wesley C. Warren, Elaine R. Mardis, George M. Weinstock, Richard K. Wilson, Genome Institute at Washington University, Ewan Birney, Elliott H. Margulies, Javier Herrero, Eric D. Green, David Haussler, Adam Siepel, Nick Goldman, Katherine S. Pollard, Jakob S. Pedersen, Eric S. Lander & Manolis Kellis

| | | | | | |
|---|---|---|---|---|---|
| Beijing Genomics Institute | 2015 | Dolphin raw sequence reads | https://www.ncbi.nlm.nih.gov/sra/SRX1136398 | Publicly available at the NCBI Sequence Read Archive (accession no. SRX1136398) |
| Beijing Genomics Institute | 2015 | Dolphin raw sequence reads | https://www.ncbi.nlm.nih.gov/sra/SRX1136399 | Publicly available at the NCBI Sequence Read Archive (accession no. SRX1136399) |
| Beijing Genomics Institute | 2015 | Dolphin raw sequence reads | https://www.ncbi.nlm.nih.gov/sra/SRX1136400 | Publicly available at the NCBI Sequence Read Archive (accession no. SRX1136400) |
| The National Institute of Standards and Technology | 2016 | Dolphin raw sequence reads | https://www.ncbi.nlm.nih.gov/sra/SRX2439826 | Publicly available at the NCBI Sequence Read Archive (accession no. SRX2439826) |

| | | | | |
|---|---|---|---|---|
| The National Institute of Standards and Technology | 2016 | Dolphin raw sequence reads | https://www.ncbi.nlm.nih.gov/sra/SRX2439828 | Publicly available at the NCBI Sequence Read Archive (accession no. SRX2439828) |
| The National Institute of Standards and Technology | 2016 | Dolphin raw sequence reads | https://www.ncbi.nlm.nih.gov/sra/SRX2439829 | Publicly available at the NCBI Sequence Read Archive (accession no. SRX2439829) |
| Birney, Elliott H. Margulies, Javier Herrero, Eric D. Green, David Haussler, Adam Siepel, Nick Goldman, Katherine S. Pollard, Jakob S. Pedersen, Eric S. Lander & Manolis Kellis | 2011 | Dolphin raw sequence reads | ftp://ftp-private.ncbi.nlm.nih.gov/pub/TraceDB/tursiops_truncatus | Available to download from the NCBI FTP site |
| Foote AD, Liu Y, Thomas GW, Vinař T, Alföldi J, Deng J, Dugan S, van Elk CE, Hunter ME, Joshi V, Khan Z, Kovar C, Lee SL, Lindblad-Toh K, Mancia A, Nielsen R, Qin X, Qu J, Raney BJ, Vijay N, Wolf JB, Hahn MW, Muzny DM, Worley KC, Gilbert MT, Gibbs RA. | 2015 | Killer whale raw sequence reads | https://www.ncbi.nlm.nih.gov/sra/SRX188930 | Publicly available at the NCBI Sequence Read Archive (accession no. SRX188930) |
| Foote AD, Liu Y, Thomas GW, Vinař T, Alföldi J, Deng J, Dugan S, van Elk CE, Hunter ME, Joshi V, Khan Z, Kovar C, Lee SL, Lindblad-Toh K, Mancia A, Nielsen R, Qin X, Qu J, Raney BJ, Vijay N, Wolf JB, Hahn MW, Muzny DM, Worley KC, Gilbert MT, Gibbs RA. | 2015 | Killer whale raw sequence reads | https://www.ncbi.nlm.nih.gov/sra/SRX188933 | Publicly available at the NCBI Sequence Read Archive (accession no. SRX188933) |
| Hyung-Soon Yim, Yun Sung Cho, Xuanmin Guang, Sung Gyun Kang, Jae-Yeon Jeong, Sun-Shin Cha, Hyun-Myung Oh, Jae-Hak Lee, Eun Chan Yang, Kae Kyoung Kwon, Yun Jae Kim, Tae Wan Kim, Wonduck Kim, Jeong Ho Jeon, Sang-Jin Kim, Dong Han Choi, Sungwoong Jho, Hak-Min Kim, Junsu Ko, Hyunmin Kim, Young-Ah Shin, Hyun-Ju Jung, Yuan | 2014 | Minke whale raw sequence reads | https://www.ncbi.nlm.nih.gov/sra/SRX302112 | Publicly available at the NCBI Sequence Read Archive (accession no. SRX302112) |

| | | | | |
|---|---|---|---|---|
| Zheng, Zhuo Wang, Yan Chen, Ming Chen, Awei Jiang, Erli Li, Shu Zhang, Haolong Hou, Tae Hyung Kim, Lili Yu, Sha Liu, Kung Ahn, Jesse Cooper, Sin-Gi Park, Chang Pyo Hong, Wook Jin, Heui-Soo Kim, Chankyu Park, Kyooyeol Lee, Sung Chun, Phillip A Morin, Stephen J O'Brien, Hang Lee, Jumpei Kimura, Dae Yeon Moon, Andrea Manica, Jeremy Edwards, Byung Chul Kim, Sangsoo Kim, Jun Wang, Jong Bhak, Hyun Sook Lee & Jung-Hyun Lee | | | | |
| Hyung-Soon Yim, Yun Sung Cho, Xuanmin Guang, Sung Gyun Kang, Jae-Yeon Jeong, Sun-Shin Cha, Hyun-Myung Oh, Jae-Hak Lee, Eun Chan Yang, Kae Kyoung Kwon, Yun Jae Kim, Tae Wan Kim, Wonduck Kim, Jeong Ho Jeon, Sang-Jin Kim, Dong Han Choi, Sung-woong Jho, Hak-Min Kim, Junsu Ko, Hyunmin Kim, Young-Ah Shin, Hyun-Ju Jung, Yuan Zheng, Zhuo Wang, Yan Chen, Ming Chen, Awei Jiang, Erli Li, Shu Zhang, Haolong Hou, Tae Hyung Kim, Lili Yu, Sha Liu, Kung Ahn, Jesse Cooper, Sin-Gi Park, Chang Pyo Hong, Wook Jin, Heui-Soo Kim, Chankyu Park, Kyooyeol Lee, Sung Chun, Phillip A Morin, Stephen J O'Brien, Hang Lee, Jumpei Kimura, Dae Yeon Moon, Andrea Manica, Jeremy Edwards, Byung Chul Kim, Sangsoo Kim, Jun Wang, Jong Bhak, Hyun Sook Lee & Jung-Hyun Lee | 2015 | Minke whale raw sequence reads | https://www.ncbi.nlm.nih.gov/sra/SRX316745 | Publicly available at the NCBI Sequence Read Archive (accession no. SRX316745) |
| Ketil Malde, Bjør-ghild B. Seliussen, María Quintela, | 2016 | Minke whale raw sequence reads | https://www.ncbi.nlm.nih.gov/sra/SRX2007381 | Publicly available at the NCBI Sequence Read Archive |

| | | | | |
|---|---|---|---|---|
| Geir Dahle, Francois Besnier, Hans J. Skaug, Nils Øien, Hiroko K. Solvang, Tore Haug, Rasmus Skern-Mauritzen, Naohisa Kanda, Luis A. Pastene, Inge Jonassen and Kevin A. Glover | | | | (accession no. SRX200 7381) |
| Ketil Malde, Bjørghild B. Seliussen, María Quintela, Geir Dahle, Francois Besnier, Hans J. Skaug, Nils Øien, Hiroko K. Solvang, Tore Haug, Rasmus Skern-Mauritzen, Naohisa Kanda, Luis A. Pastene, Inge Jonassen and Kevin A. Glover | 2016 | Minke whale sequence reads | https://www.ncbi.nlm. nih.gov/sra/SRX2007379 | Publicly available at the NCBI Sequence Read Archive (accession no. SRX200 7379) |
| Ketil Malde, Bjørghild B. Seliussen, María Quintela, Geir Dahle, Francois Besnier, Hans J. Skaug, Nils Øien, Hiroko K. Solvang, Tore Haug, Rasmus Skern-Mauritzen, Naohisa Kanda, Luis A. Pastene, Inge Jonassen and Kevin A. Glover | 2016 | Minke whale sequence reads archive | https://www.ncbi.nlm. nih.gov/sra/SRX2007373 | Publicly available at the NCBI Read Archive (accession no. SRX2007373) |
| Ketil Malde, Bjørghild B. Seliussen, María Quintela, Geir Dahle, Francois Besnier, Hans J. Skaug, Nils Øien, Hiroko K. Solvang, Tore Haug, Rasmus Skern-Mauritzen, Naohisa Kanda, Luis A. Pastene, Inge Jonassen and Kevin A. Glover\ | 2016 | Minke whale sequence reads | https://www.ncbi.nlm. nih.gov/sra/SRX2007368 | Publicly available at the NCBI Sequence Read Archive (accession no. SRX200 7368) |
| Wesley C. Warren, Lukas Kuderna, Alana Alexander, Julian Catchen, Jose G. Perez-Silva, Carlos Lopez-Otın, Vıctor Quesada, Patrick Minx, Chad Tomlinson, Michael J. Montague, Fabiana H. G. Farias, Ronald B. Walter, Tomas Marques-Bonet, Travis Glenn, Troy J. Kieran, Sandra S. Wise, John Pierce Wise Jr, Robert M. Waterhouse, and John Pierce Wise Sr | 2017 | Sperm whale sequence reads | https://www.ncbi.nlm. nih.gov/sra/SRX220365 | Publicly available at the NCBI Sequence Read Archive (accession no. SRX220 365) |
| Wesley C. Warren, Lukas Kuderna, | 2017 | Sperm whale raw sequence reads | https://www.ncbi.nlm. nih.gov/sra/SRX220366 | Publicly available at the NCBI Sequence |

| Author(s) | Year | Dataset title | URL | Database, license, and accessibility information |
|---|---|---|---|---|
| Alana Alexander, Julian Catchen, Jose G. Perez-Silva, Carlos Lopez-Otın, Vıctor Quesada, Patrick Minx, Chad Tomlinson, Michael J. Montague, Fabiana H. G. Farias, Ronald B. Walter, Tomas Marques-Bonet, Travis Glenn, Troy J. Kieran, Sandra S. Wise, John Pierce Wise Jr, Robert M. Waterhouse, and John Pierce Wise Sr | | | | Read Archive (accession no. SRX220 366) |
| Wesley C. Warren, Lukas Kuderna, Alana Alexander, Julian Catchen, Jose G. Perez-Silva, Carlos Lopez-Otın, Vıctor Quesada, Patrick Minx, Chad Tomlinson, Michael J. Montague, Fabiana H. G. Farias, Ronald B. Walter, Tomas Marques-Bonet, Travis Glenn, Troy J. Kieran, Sandra S. Wise, John Pierce Wise Jr, Robert M. Waterhouse, and John Pierce Wise Sr | 2017 | Sperm whale sequence reads | https://www.ncbi.nlm.nih.gov/sra/SRX220367 | Publicly available at the NCBI Sequence Read Archive (accession no. SRX220 367) |
| Puli Chandramouli Reddy, Ishani Sinha, Ashwin Kelkar, Farhat Habib, Saurabh J. Pradhan, Raman Sukumar, Sanjeev Galande | 2015 | Asian elephant raw sequence reads | https://www.ncbi.nlm.nih.gov/sra/SRX1427123 | Publicly available at the NCBI Sequence Read Archive (accession no. SRX1427123) |
| Palkopoulou E, Lipson M, Mallick S, Nielsen S, Rohland N, Baleka S, Karpinski E, Ivancevic AM, To TH, Kortschak RD, Raison JM, Qu Z, Chin TJ, Alt KW, Claesson S, Dalén L, MacPhee RDE, Meller H, Roca AL, Ryder OA, Heiman D, Young S, Breen M, Williams C, Aken BL, Ruffier M, Karlsson E, Johnson J, Di Palma F, Alfoldi J, Adelson DL, Mailund T, Munch K, Lindblad-Toh K, Hofreiter M, Poinar H, Reich D | 2018 | Forest elephant raw sequence reads | https://www.ebi.ac.uk/ena/data/view/ERR2260500 | Publicly available at the NCBI Sequence Read Archive (accession no. ERR2260500) |
| Palkopoulou E, Lipson M, Mallick S, Nielsen S, Rohland N, Baleka S, Kar- | 2018 | Forest elephant raw sequence reads | https://www.ebi.ac.uk/ena/data/view/ERR2260495 | Publicly available at the European Nucleotide Archive (accession no. |

| | | | | |
|---|---|---|---|---|
| pinski E, Ivancevic AM, To TH, Kortschak RD, Raison JM, Qu Z, Chin TJ, Alt KW, Claesson S, Dalén L, MacPhee RDE, Meller H, Roca AL, Ryder OA, Heiman D, Young S, Breen M, Williams C, Aken BL, Ruffier M, Karlsson E, Johnson J, Di Palma F, Alfoldi J, Adelson DL, Mailund T, Munch K, Lindblad-Toh K, Hofreiter M, Poinar H, Reich D | | | | ERR2260495) |
| Diego Cortez, Ray Marin, Deborah Toledo-Flores, Laure Froidevaux, Angélica Liechti, Paul D. Waters, Frank Grützner & Henrik Kaessmann | 2014 | Savanna elephant raw sequence reads | https://www.ncbi.nlm.nih.gov/sra/SRX339471 | Publicly available at the NCBI Sequence Read Archive (accession no. SRX339471) |
| Diego Cortez, Ray Marin, Deborah Toledo-Flores, Laure Froidevaux, Angélica Liechti, Paul D. Waters, Frank Grützner & Henrik Kaessmann | 2014 | Savanna elephant raw sequence reads | https://www.ncbi.nlm.nih.gov/sra/SRX339470 | Publicly available at the NCBI Sequence Read Archive (accession no. SRX339470) |
| Palkopoulou E, Lipson M, Mallick S, Nielsen S, Rohland N, Baleka S, Karpinski E, Ivancevic AM, To TH, Kortschak RD, Raison JM, Qu Z, Chin TJ, Alt KW, Claesson S, Dalén L, MacPhee RDE, Meller H, Roca AL, Ryder OA, Heiman D, Young S, Breen M, Williams C, Aken BL, Ruffier M, Karlsson E, Johnson J, Di Palma F, Alfoldi J, Adelson DL, Mailund T, Munch K, Lindblad-Toh K, Hofreiter M, Poinar H, Reich D | 2018 | American mastodon raw sequence reads | https://www.ebi.ac.uk/ena/data/view/ERR2260508 | Publicly available at the European Nucleotide Archive (accession no. ERR2260508) |
| Palkopoulou E, Lipson M, Mallick S, Nielsen S, Rohland N, Baleka S, Karpinski E, Ivancevic AM, To TH, Kortschak RD, Raison JM, Qu Z, Chin TJ, Alt KW, Claesson S, Dalén L, MacPhee RDE, Meller H, Roca AL, Ryder OA, Heiman D, Young S, Breen | 2018 | American mastodon raw sequence reads | https://www.ebi.ac.uk/ena/data/view/ERR2260503 | Publicly available at the European Nucleotide Archive (accession no. ERR2260503) |

| | | | | |
|---|---|---|---|---|
| M, Williams C, Aken BL, Ruffier M, Karlsson E, Johnson J, Di Palma F, Alfoldi J, Adelson DL, Mailund T, Munch K, Lindblad-Toh K, Hofreiter M, Poinar H, Reich D | | | | |
| Hyung-Soon Yim, Yun Sung Cho, Xuanmin Guang, Sung Gyun Kang, Jae-Yeon Jeong, Sun-Shin Cha, Hyun-Myung Oh, Jae-Hak Lee, Eun Chan Yang, Kae Kyoung Kwon, Yun Jae Kim, Tae Wan Kim, Wonduck Kim, Jeong Ho Jeon, Sang-Jin Kim, Dong Han Choi, Sung-woong Jho, Hak-Min Kim, Junsu Ko, Hyunmin Kim, Young-Ah Shin, Hyun-Ju Jung, Yuan Zheng, Zhuo Wang, Yan Chen, Ming Chen, Awei Jiang, Erli Li, Shu Zhang, Haolong Hou, Tae Hyung Kim, Lili Yu, Sha Liu, Kung Ahn, Jesse Cooper, Sin-Gi Park, Chang Pyo Hong, Wook Jin, Heui-Soo Kim, Chankyu Park, Kyooyeol Lee, Sung Chun, Phillip A Morin, Stephen J O'Brien, Hang Lee, Jumpei Kimura, Dae Yeon Moon, Andrea Manica, Jeremy Edwards, Byung Chul Kim, Sangsoo Kim, Jun Wang, Jong Bhak, Hyun Sook Lee & Jung-Hyun Lee | 2016 | Minke whale raw sequence reads | https://www.ncbi.nlm.nih.gov/sra/SRX318063 | Publicly available at the Sequence Read Archive (accession no. SRX318063) |

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
