## [Decision Letter]

Thank you for submitting your article "Recurrent loss of *HMGCS2* shows that ketogenesis is not essential for the evolution of large mammalian brains" for consideration by *eLife*. Your article has been reviewed by three peer reviewers, including Vincent J Lynch as a guest Reviewing Editor and Reviewer #1, and the evaluation has been overseen by Wittkopp as the Senior Editor. The following individuals involved in review of your submission have agreed to reveal their identity: Cristian Cañestro (Reviewer #2); Kevin Campbell (Reviewer #3).

The reviewers have discussed the reviews with one another and the Reviewing Editor has drafted this decision to help you prepare a revised submission.

Summary:

In this manuscript Jebb and Hiller report the loss of the *HMGCS2* gene in three mammalian lineages, which is very surprising given the role *HMGCS2* plays in acetoacetate synthesis. This is surprising because fatty acid-derived ketone bodies provide energy for the brain during neonatal development and fasting suggesting there is strong selection to maintain this gene and pathway. These data suggest that cetaceans, Old World fruit bats, and Proboscideans either evolved alternative metabolic strategies to deal with *HMGCS2* loss or some other functionally similar gene performs a similar role as *HMGCS2*.

Essential revisions:

1) If ketones are no longer used for ATP production do other genes in the pathway show evidence of reduced purifying selection or loss?

2) Given the ability to estimate a relative date of loss of elephants why not bats and cetaceans?

3) Please confirm *HMGCS2* loss using the more recent loxAfr4 assembly; It's likely missing in loxAfr4 as well but best to confirm.

4) The fact that HMGCS1 is localized in the cytosol, and not in mitochondria, implies that HMGCS1 cannot rescue the lack of HMCGS2 (if so, please mention in the manuscript, and provide references supporting this fact). Along these lines, is it possible to show that HMGCS1 does not have a putative gain of a "novel alternative exon1" in HMGCS1 coding for a mitochondrial targeting domain could allow to make this paralog to be functionally redundant with *HMGCS2* (and making it therefore dispensable)? While the reviewers understand that this scenario may be remote, in such case those species could continue producing ketone bodies, which therefore would significantly affect the main conclusions of the manuscript. Is there EST (transcriptomic) data for any of the species that has lost HMCGS2 to discard (or at least check) that possibility? Gene prediction software could also be helpful to check for the presence of such potential mitochondria-target exon in HMGCS1 (which in principle should be shared within each group of mammals that share the loss to corroborate its presence).

---

## [Author Response]

Essential revisions:1) If ketones are no longer used for ATP production do other genes in the pathway show evidence of reduced purifying selection or loss?

Thank you for raising this question. The first and third steps in the ketogenesis pathway are catalyzed by *ACAT1* and *HMGCL*, respectively. We found that these genes do not have any inactivating mutations in the lineages that lost *HMGCS2*. This is probably explained by the fact that both enzymes are also required for amino acid metabolism. *BDH1* is a non-pleiotropic enzyme that converts the ketone body acetoacetate into the ketone body d-β-hydroxybutyrate. We found that this gene is lost in cetaceans and pteropodids, where several inactivating mutations are present together with a signature of relaxed selection. No evidence for loss or relaxed selection was found in the elephant. Analyzing this gene in other mammals, we found that the shrew probably also lacks this enzyme, while *ACAT1, HMGCS2* and *HMGCL* are fully intact in this species. Similar to *Bdh1* knockout mice, acetoacetate and acetone are likely the main ketone bodies in the shrew.

These findings are shown in a new Figure 1—figure supplement 5 and described in a new paragraph:

“Next, we investigated whether the loss of *HMGCS2* is associated with the loss of other enzymes in the ketogenesis pathway (Figure 1A). […] Overall, this suggests that the loss of *HMGCS2* is only associated with the loss of non-pleiotropic genes in the ketogenesis pathway.”

We also added the three gene symbols to Figure 1A and better described enzyme function in the legend:

“With the exception of the mitochondrial HMG-CoA synthase (*HMGCS2*, red font), the other two enzymes required for acetoacetate production also have roles in amino acid metabolism and are thus pleiotropic. *BDH1* is only required for converting acetoacetate into d-β-hydroxybutyrate.

2) Given the ability to estimate a relative date of loss of elephants why not bats and cetaceans?

As suggested, we have now performed molecular dating for the fruit bat and cetacean lineage. To get estimates as precise as possible, we first investigated *HMGCS2* in species that represent their closest outgroups, using new genomes that recently became available. For the fruit bats, we inspected the genome of the horseshoe bat (the closest insectivorous outgroup) and found that this bat has an intact, annotated gene. For cetaceans, we aligned the hippo genome, which also revealed an intact gene. Both species have been added to the phylogenetic tree in Figure 1B. To estimate a Ka/Ks value for the internal branches along which the gene was lost, we used PAML. For the fruit bats, we estimate that *HMGCS2* loss occurred rather late on the branch and overlaps the split of the flying foxes and the Egyptian fruit bat (note that it cannot be resolved whether the gene was lost before or after the split since the gene is entirely deleted in the latter species). For cetaceans, we estimate a loss relatively soon after the split from the hippo lineage. To present these results, we decided to split the previous Figure 1 and show the molecular dating results for all three lineages in a new Figure 2A-C.

Since our full alignment now contains two additional species, we re-estimated the date of *HMGCS2* loss in the elephant lineage with PAML. Furthermore, we now included not only the African savanna elephant sequence in the alignment but also the sequences from the forest elephant, Asian elephant and mastodon that we obtained by aligning genomic read data. While this makes the assumption that the other Elephantimorpha species share the same deletion of *HMGCS2* exons 1-5, we believe adding additional species will increase the precision of the estimate. As shown in Supplementary file 3, with additional Elephantimorpha, we obtained a younger, probably more precise loss estimate of 45 to 42 Mya, which overlaps the split of the low-EQ Moeritherium. Importantly, the Palaeomastodon, a more recent lineage that split ~37 Mya from other proboscids (added to Figure 2C), is also described to have had a small braincase and thus probably had a low EQ, which suggests that loss of ketogenesis still predates the period of brain size expansion in Proboscidea.

3) Please confirm HMGCS2 loss using the more recent loxAfr4 assembly; It's likely missing in loxAfr4 as well but best to confirm.

We downloaded the loxAfr4 genome and aligned it to the human genome. The loxAfr4 genome assembly confirms the genomic rearrangements including the deletion of most *HMGCS2* exons. We added to the legend of Figure 1—figure supplement 2:

“Alignment chains to the African savanna elephant loxAfr4 assembly are virtually identical and confirm the partial *HMGCS2* deletion.”

4) The fact that HMGCS1 is localized in the cytosol, and not in mitochondria, implies that HMGCS1 cannot rescue the lack of HMCGS2 (if so, please mention in the manuscript, and provide references supporting this fact). Along these lines, is it possible to show that HMGCS1 does not have a putative gain of a "novel alternative exon1" in HMGCS1 coding for a mitochondrial targeting domain could allow to make this paralog to be functionally redundant with HMGCS2 (and making it therefore dispensable)? While the reviewers understand that this scenario may be remote, in such case those species could continue producing ketone bodies, which therefore would significantly affect the main conclusions of the manuscript. Is there EST (transcriptomic) data for any of the species that has lost HMCGS2 to discard (or at least check) that possibility? Gene prediction software could also be helpful to check for the presence of such potential mitochondria-target exon in HMGCS1 (which in principle should be shared within each group of mammals that share the loss to corroborate its presence).

We investigated the *HMGCS1* protein sequences in each loss species with TargetP, which reveals no evidence for the presence of a mitochondrial targeting domain. In order to identify novel or alternative first exons that could provide a new N-terminus, we first inspected available Augustus gene predictions for putative alternative *HMGCS1* transcripts and again found no mitochondrial targeting domain. Second, for the Egyptian fruit bat and the minke whale, RNA-seq data from the liver (the site of ketogenesis in other species) is available. By processing and analyzing this data, we found no evidence for alternative or novel first exons in the Egyptian fruit bat. For the minke whale, RNA-seq data showed an upstream first exon; however, this exon did not contain an ATG start codon, showing that this exon is part of the UTR and that the minke whale *HMGCS1* does not encode a different N-terminus.

These results are added as to the main text:

“Finally, we considered the possibility that *HMGCS1*, the cytosolic HMG-CoA synthase, may compensate for *HMGCS2* loss, which would require *HMGCS1* to be localized in the mitochondria, where ketogenesis happens in other species. […] Thus, HMGCS1 does not seem to be capable of compensating for the loss of *HMGCS2*, suggesting that ketogenesis is lost in cetaceans, pteropodids and Elephantimorpha.”

And as a new section to the Materials and methods:

“Investigating the possibility of co-option of *HMGCS1*

We tested the amino acid sequences of the annotated or CESAR-inferred *HMGCS1* protein from all *HMGCS2* loss species for the presence of a potential mitochondrial target peptide (mTP) using TargetP [Emanuelsson et al., 2007]. […] For both species, we found no evidence of alternative or novel exons that could result in a different HMGCS1 N-terminus.”